# An Alternative Study about the Geometry and the First Law of Thermodynamics for AdS Lovelock Gravity, Using the Definition of Conserved Charges

**DOI:** 10.3390/e24091197

**Published:** 2022-08-27

**Authors:** Rodrigo Aros, Milko Estrada, Pablo Pereira

**Affiliations:** 1Departamento de Ciencias Fisicas, Universidad Andres Bello, Av. Republica 252, Santiago 8370134, Chile; 2Facultad de Ingeniería, Ciencia y Tecnología, Universidad Bernardo O’Higgins, Av. Viel 1497, Santiago 8370993, Chile; 3Departamento de Fisica, Universidad del Biobio, Av. Collao 1202, Concepcion 4051381, Chile

**Keywords:** black holes thermodynamics, higher curvature gravity, higher dimensional gravity

## Abstract

In this work, we introduce an extension of the study of the first law of thermodynamics of black holes based on the geometry of the extended phase space for AdS Lovelock gravities, which includes changes in scales. As expected, the result obtained coincides with the previously known four-dimensional case. For higher dimensions, the result is the rise of two new contributions to the first law of thermodynamics. The first term corresponds to corrections of the usual definition of thermodynamic volumes at the horizon due to the presence of the higher curvature terms. The second term arises in odd dimensions, comes from the asymptotic region, and corresponds to a scale transformation of the form $\propto \hat{\delta }\ln\left(l/\ell \right)$, with *l* the AdS radius and *ℓ* a parameter. A particularly interesting case corresponds to the Chern Simons gravity where the change scale does not generate a contribution at the asymptotic region, likely due to the Chern Simons AdS local symmetry.

## 1. Introduction

The discovery that accretion processes around a black hole can be reinterpreted as thermodynamic processes was one of the greatest breakthroughs in theoretical physics. In this respect, it is worth mentioning that in reference [1], it was shown that the entropy production itself can be viewed as a Noether–conserved quantity, which certainly could be relevant to black hole accretion processes.

The original derivation by Carter [2], Bardeen [3], Bekenstein [4], Hawking [5], and many others, was based, roughly speaking, on the idea that in-falling matter, in pseudo-adiabatic processes, introduces small perturbations on the black hole that can be expressed as infinitesimal changes in the space of parameters that characterize the black hole solution. Given that the black hole must evolve into another black hole solution, the variation of those changes is constrained by the *first law of the black hole thermodynamics*, i.e., by a law of the form:(1)δM=TδS+…,
where—for a black hole—one can define a temperature *T* and an entropy *S*. Although heretofore, there is no agreement on which *micro-states* give rise to this entropy, still these results are widely accepted. In many ways, this can be considered the starting point upon later, the holographic principle, originally proposed by t’Hooft and Susskind [6,7,8], was constructed.

Although the many derivations of Equation (Equation 1) are expected to be connected, in one way or another, there is no certainty of this [9]. In the case of asymptotically locally AdS, one can refer to [10], where different approaches to define conserved charges are discussed. Therefore, the analysis of the first law of thermodynamics by means of conserved charges is still an open question.

One can notice in Equation (Equation 1) the lack of the *work* term, −Pδv. Obviously, to include such a term requires introducing definitions for both pressure and volume. This has been conducted in many different ways, but there is a general agreement that it is necessary to promote the mass parameter, *M* in Equation (Equation 1), from the energy of the system to its enthalpy, *H*, such as the first law of *black hole* thermodynamics adopt the form:(2)$$\delta H=\delta M=T\delta S+\Omega_{h}\delta J+\Phi \delta Q+V\delta P,$$
where *V* is the function of the black hole radius $r_{+}$. In [11], for four dimensions, it was proposed that the thermodynamics pressure satisfies P∼−Λ with Λ, the cosmological constant. See [12,13] for different discussions on this.

The introduction of pressure *P* as a new thermodynamic variable in black hole thermodynamics determines what is called an extended phase space. However, as occurred with the standard BH thermodynamics, this is a broad name that includes many different derivations, as mentioned above. Because of that, many interesting results of these improved thermodynamics have been obtained during the last decade. In reference [14], it was shown that some charged black holes exhibit a P−V critical behavior where the small/large black hole phase transitions are analogous to the liquid/gas phase transitions in a Van der Waals fluid. Regarding this, in [15], the Hawking Page phase transition was restudied. In [16], the Van der Waals behavior, re-entrant phase transitions, and tricritical points were tested. Other examples of P−V critical behavior were discussed in [17,18,19,20,21,22].

### 1.1. Change of the Cosmological Constant in Four Dimensions and Generalizations

As a starting point in the discussion, let us review the consequences of considering changes in the cosmological constant in four dimensions. The simplest case where this can be realized is the four-dimensional Schwarzschild–AdS spaces. In Schwarzschild coordinates, the line element reads:(3)$$ds^{2}=-f{\left(r\right)}^{2}dt^{2}+\frac{1}{f{\left(r\right)}^{2}}dr^{2}+r^{2}\left(d\theta^{2}+\sin{\left(\theta \right)}^{2}d\phi^{2}\right)$$
with:$$f{\left(r\right)}^{2}=1+\frac{r^{2}}{l^{2}}-2\frac{M}{r}.$$
Here, the cosmological constant is given by $-3l^{-2}$. Remarkably, one can interpret $f(r_{+})=0$ as a curve in a space defined by the *coordinates*$\left(r_{+},M,l\right)$ and from this to study the thermodynamics of this solution. This is called the extended phase space.

It is obvious that any modification of the cosmological constant, due to $\Lambda =-3l^{-2}$, can be reinterpreted as a change of the *AdS radius*$l^{2}$ of the geometry. This transformation, in turn, can be promoted to a change of scale of the geometry. To observe this, let us consider $\Lambda =-3l^{-2}$, and the transformation l→(1+σ)l=l+δl, with |σ|≪1. This transformation can rewritten as:(4)$$-2\Lambda \sqrt{\det g}d^{4}x=\frac{6}{l^{2}}\sqrt{\det g}d^{4}x\Rightarrow \frac{6}{l^{2}}\left(1-2\sigma \right)\sqrt{\det g}d^{4}x=\frac{6}{l^{2}}\sqrt{\det\tilde{g}}d^{4}x$$
where ${\tilde{g}}_{\mu \nu }=\left(1-\sigma \right)g_{\mu \nu }$. This corresponds to a rigid Weyl transformation.

A consequence of the previous rigid transformation of scale at the bulk is the induction of a Weyl transformation at the conformal infinity [23,24,25]. For a review of conformal transformations in this context, see [26]. This has some interesting consequences in the context of the AdS/CFT, as discussed in [27].

Now, one can notice that most of the ideas mentioned above are not restricted to only Einstein gravity or four dimensions. To address the problem in higher dimensions, we will study only Lovelock gravity because this preserves the same features of GR in four dimensions [28].

To begin with a discussion, one can notice that in four dimensions, in the absence of matter, there are only two coupling constants. This scenario changes in higher dimensions where additional terms can be incorporated in the Lagrangians, and with them, additional coupling constants arise. In principle, for each of these new coupling constants, one could introduce an additional extra dimension in the space of the parameters mentioned above. Unlike four dimensions moving in that enlarged space of parameters, we can modify the asymptotia of the solutions. For Lovelock gravity, this can be confirmed as follows. Lovelock gravity has different *k*-fold degenerated ground states, each being a manifold of constant curvature $\lambda_{i}^{2}$ with $\lambda_{i}$ and *k* defined by the coupling constants in the Lagrangian; see below. Evidently, if the coupling constants vary independently, not only the values of $\lambda_{i}^{2}$ would change, but also the degeneration of each of the ground states *k* and also the asymptotia of the non-ground state solutions. To avoid this, one can make each of the constant functionals dependent on a single scale, using the one provided by the AdS radii. This maintains the asymptotia and, at the same time, allows us to study the changes of that scale. This can be considered one of the simplest generalizations of *changing the cosmological constant in four dimensions*. Conversely, only changing the cosmological constant and not the rest of the coupling constants in the Lovelock Lagrangian accordingly, see below, would spoil the asymptotic behavior of the solutions.

### 1.2. Thermodynamics

As mentioned above, there are several approaches to construct the thermodynamics of a black hole. In this work, a generalization of Wald’s deviation [29] that considers transformation of the form (Equation 4) will be explored. In [30], a similar idea was explored in four dimensions in terms of the cosmological constant.

In the original proposal by Wald, the thermodynamics arises as a consequence of changes in parameters that define the classical solutions. In this context, to be on-shell replaces the thermal equilibrium and the variation of the parameter becomes analogous to the more standard quasi-static evolution of the thermodynamics. The variation of parameters originally discussed in [31] corresponds to the variation of the Hamiltonian charges expressed in terms of the variations of the Noether’s charges and addition boundary terms. The entropy is obtained in terms of the Noether charge associated with the Killing vector that defines the (Killing) horizon of the geometry [29].

It is worth noticing that Wald’s approach requires implementing certain considerations before the computations can be performed properly. For asymptotic flat spaces, the process of defining conserved charge can be cumbersome but it is usually free of divergences. However, the AdS asymptotia requires the introduction of a regularization process to define the Noether’s charges computed at infinity. To our knowledge, in doing this, any method defines charges that are independent of AdS radius *l* (or the cosmological constant). The consequences of this are two-fold. On one hand, this seems to introduce to a sort regulator or a naïve regularization parameter into the computations defined as r/l with *r*, a radial coordinate. On the other hand, the independence on *l*, or equivalently on the cosmological constant, of the conserved charges is usually considered a hint of a conformal pedigree of these charges, in spite of the gravitational theory is not conformal. This last idea is reinforced by the fact that it is direct to check that the Kerr–Newman–AdS thermodynamics behaves as the thermodynamics of a conformal theory, see, for instance [32]. See [33], about the correlation between the emission modes and temperature of the event horizon in Einstein Gauss Bonnnet gravity.

Unlike the charges computed at (the conformal) infinity, the charges computed at the horizon are affected by any change of scale. This naturally implies that additional terms must arise to compensate the transformation of scale if a thermodynamic relation holds. This must be valid for any method to obtain the thermodynamics, including those mentioned above and the Hamiltonian approach [34]. The additional terms for Einstein gravity in four dimensions [35] is given by:(5)$$v\delta P=\frac{4}{3}\pi r_{+}^{3}\delta \left(\frac{1}{l^{2}}\right).$$
where $P=l^{-2}$ and $V=\frac{4}{3}\pi r_{+}^{3}$ coincide with the naive definition of volume of a black hole whose radius is $r_{+}$. This is not the geometric volume.

Therefore, it is of physical interest to explore the physical consequences in the first law of thermodynamics of incorporating changes of scales (which preserve the asymptotic structure), and testing the contributions of the Noether charge to the first law, both at the horizon, as the asymptotic region.

In the next sections, this problem will be extended in general terms to any asymptotically local AdS solutions of Lovelock gravity. The fundamental result is the definition of thermodynamic volumes which depend of the theory and a universal expression for the thermodynamics pressure given by $P∼l^{-2}$. This parallels the dependencies of the Entropy and Temperature, respectively, in the sense that the temperature also has a purely geometrical origin, while the expression of entropy depends on the theory considered [29,36]. To do this, an extension of the formalism developed in [31] that incorporates changes of scales that preserve the asymptotic structure is constructed. For simplicity, the computations will be carried out in first order formalism of gravity. The connection with the second order formalism (metric formalism) will be explored elsewhere.

## 2. Phase Space and Charges

### 2.1. Noether Charges

Let us start by rephrasing the construction of the Noether currents associated with symmetry. In general, the most general infinitesimal transformation of a field ϕ(x) is given as:(6)$$x\to x^{\prime }=x+\xi \left(x\right)\mathrm{and}\phi \left(x\right)\to \phi^{\prime }\left(x^{\prime }\right).$$
Now, the infinitesimal transformation, defined as $\delta \phi =\phi^{\prime }\left(x^{\prime }\right)-\phi \left(x\right)$, can be split into $\delta \phi =\phi^{\prime }\left(x^{\prime }\right)-\phi \left(x^{\prime }\right)+\phi \left(x^{\prime }\right)-\phi \left(x\right)$. Here, one can recognize the usual function variation $\delta_{0}\phi =\phi^{\prime }\left(x^{\prime }\right)-\phi \left(x^{\prime }\right)$ and the Lie derivative, $\phi \left(x^{\prime }\right)-\phi \left(x\right)={𝓛}_{\xi }\phi$, along the diffeomorphism defined by ξ(x).

Now, a transformation that defines a symmetry of an action principle is:(7)$$I=\int_{M_{d}}L\left(\phi \right)$$
where L is d−form Lagrangian, provided L(ϕ) and L(ϕ+δϕ) have the same equations of motion (EOM). This can be written formally in terms of the transformations as:(8)$$\delta L\left(\phi \right)=\delta_{0}L\left(\phi \right)+{𝓛}_{\xi }L\left(\phi \right)=d\Psi .$$
where:(9)$$\delta_{0}L\left(\phi \right)={\mathrm{EOM}}_{\phi }\delta_{0}\phi +d\Theta \left(\delta_{0}\phi ,\phi \right).$$
where ${\mathrm{EOM}}_{\phi }$ stands for the equations of motion associated with ϕ. Here, $\Theta (\delta_{0}\phi ,\phi )$ is called the *boundary term* and it is worth stressing that in order to have a proper action principle, $\Theta (\delta_{0}\phi ,\phi )$ must vanish on the boundary conditions. Finally, it is worth recalling that ${𝓛}_{\xi }L=dI_{\xi }L$ since dL≡0. Therefore, for any symmetry transformation it is possible to define:(10)$$d\left(\Theta +I_{\xi }L\left(\phi \right)-\Psi \right)=-{\mathrm{EOM}}_{\phi }\delta_{0}\phi .$$
With this in mind, one can define the n−1-form current,
(11)$${}^{*}J=\Theta \left(\delta_{0}\phi ,\phi \right)+I_{\xi }L\left(\phi \right)-\Psi$$
whose divergence vanishes on the shell, i.e., ${\left.d^{*}J\right|}_{\mathrm{On}\mathrm{Shell}}=0$. This is called the Noether current. This implies that at least locally, ${}^{*}J=dQ$. In the next section, the exact form of this current will be discussed for Lovelock gravity [25].

The definition of Noether charge ${}^{*}J$ is just a first step to define a conserved charge. In fact, to compute from Equation (Equation 11) a conserved charge is necessary to impose at least two additional conditions. First, the manifold $M_{d}$ must have at least an asymptotic time-like Killing symmetry. For simplicity, one can consider a stationary space $M_{d}=R⊗\Sigma_{d-1}$, where R stands for a time direction, but in general, it is only necessary that $R⊗\partial \Sigma_{\infty }\subset \partial M_{d}$. The second condition is that the transformation of ϕ must be defined by a (Killing) symmetry in the space of solutions, i.e., by a transformation that maps solutions into solutions. In the case of diffeomorphisms, where $\delta \phi =0=\delta_{0}\phi +{𝓛}_{\xi }\phi$, this last condition merely implies that ξ must be a Killing vector of $M_{d}$.

One aspect to consider, in addition, is that it is necessary to have a proper action principle, meaning that the action principle must be finitely evaluated on any solution that satisfies the boundary conditions and having an extreme subjected to those boundary conditions. This implies that $\Theta (\phi ,\delta_{0}\phi )$ must vanish (on shell), provided the boundary conditions are satisfied. In addition, the action must have proper conserved charges. For asymptotically AdS spaces, this implies the need to implement a regularization process on the action principle. See, for instance [32,37,38]. Essentially, the process of regularization corresponds to the addition of terms to the action principle that do not alter the EOM but satisfy the three aforementioned conditions. In order to do that, there are only two options: the addition of boundary terms Φ or the addition of topological densities $\hat{\Phi }$. This last is because any topological density, $\hat{\Phi }$, satisfies $\delta_{0}\hat{\Phi }=d\delta_{0}\Phi$. For both options, Φ must be a suitable function of the fields. Therefore, the improved action principle must be either:(12)$$L\to L^{\prime }=L+d\Phi \mathrm{or}L\to L^{\prime }=L+\hat{\Phi }.$$
The variation of the improved action principle is given by:(13)$$\delta_{0}I^{\prime }=\int_{M}\delta L^{\prime }=\int_{\partial M}\Theta \left(\delta_{0}\phi ,\phi \right)+\delta_{0}\Phi .$$
Under the suitable boundary conditions it must be satisfied that:(14)$${\left.\left(\Theta \left(\delta_{0}\phi ,\phi \right)+\delta_{0}\Phi \right)\right|}_{\partial M}=0.$$
From now on, for notation, it will be denoted as:(15)$$\Theta^{\prime }\left(\delta_{0}\phi ,\phi \right)=\Theta \left(\delta_{0}\phi ,\phi \right)+\delta_{0}\Phi .$$
It must be recalled that both the improved action principles, $I^{\prime }$, and the improved Noether charges,
(16)$${}^{*}J^{\prime }=\Theta^{\prime }\left(\delta_{0}\phi ,\phi \right)+I_{\xi }L^{\prime }\left(\phi \right)-\Psi ,$$
must be finite, in order to have a well defined action principle. This imposes strong restrictions. Fortunately, for AdS spaces, this can be accomplished. Finally, it will be denoted that the local expression of the current ${}^{*}J^{\prime }=dQ^{\prime }$. It must be stressed that $Q^{\prime }$ is connected with conserved charges only if ξ is a Killing vector.

### 2.2. The Presymplectic Form and Charges

In general, the generator, in Hamiltonian formalism, of the diffeomorphisms associated with the transformation x→x+ξ is given by:(17)$$G\left(\xi \right)=\int_{\Sigma }H_{\mu }\xi^{\mu }+\int_{\partial \Sigma }g\left(\xi \right).$$
g(ξ) is a n−2-form whose presence is necessary to construct a proper generator on the phase space of the theory [34]. The Hamiltonian charges come from this definition as the value on-shell for the Killing vector, i.e.,
(18)$$\begin{array}{ccc} {\left.G(\xi )\right|}_{\mathrm{on}-\mathrm{shell}} & = & \int_{\Sigma }{\left.\underset{=0}{\underset{︸}{H_{\mu }}}\xi^{\mu }\right|}_{\mathrm{on}-\mathrm{shell}}+\int_{\partial \Sigma }{\left.g(\xi )\right|}_{\mathrm{on}-\mathrm{shell}}\\ & = & \int_{\partial \Sigma }{\left.g(\xi )\right|}_{\mathrm{on}-\mathrm{shell}} \end{array}$$
It must be stressed that this is not modified by the presence of sources since, in general, if they are presented, they must be included in the definitions $H_{\mu }$ and g(ξ).

### 2.3. An Extended Covariant Phase Space Formalism

Before proceeding it is worth highlighting some of the elements of the original construction of the covariant phase space method [29,31]. In principle, for a given Killing vector, the corresponding Noether and Hamiltonian charges differ. Fortunately, it is possible to connect them on shell by the phase space method [31]. Let us define the d−1-form:(19)$$\chi =\delta_{1}\Theta^{\prime }\left(\phi ,\delta_{2}\phi \right)-\delta_{2}\Theta^{\prime }\left(\phi ,\delta_{1}\phi \right),$$
where $\delta_{1}$ and $\delta_{2}$ stand for transformations of the form Equation (Equation 6). As mentioned above, for simplicity, it will be considered only the stationary case $M_{d}=R\times \Sigma_{d-1}$. In addition, it must be imposed that $\partial \Sigma_{d-1}=\partial \Sigma_{\infty }⊕\partial \Sigma_{H}$ where $\partial \Sigma_{H}$ is to be connected with existence of a Killing horizon in the manifold. Under these conditions [31],
(20)$$\Xi =\int_{\Sigma }\chi =0,$$
provided either $\delta_{1}$ or $\delta_{2}$, are transformations along the space of solutions, as mentioned above.

The identity in Equation (Equation 20) contains the thermodynamics of a black hole in [29] provided one of the transformation is generated as the generator of the Killing horizon, ξ and the second one corresponds to variation along the parameters of the solution. As mentioned above, under diffeomorphisms, it is satisfied that $\delta_{0}\phi =\delta_{\xi }\phi \;=-{𝓛}_{\xi }\phi$, following [29,31], and therefore:(21)$$\chi \left(\phi ,\delta_{\xi }\phi ,\hat{\delta }\phi \right)=d\left(\hat{\delta }\left(Q^{\prime }\right)+I_{\xi }\Theta^{\prime }\left(\phi ,\hat{\delta }\phi \right)\right).$$
Finally, Equation (Equation 20) can be expressed as the *conservation relation* between the horizon and asymptotic region:(22)$$\int_{\partial \Sigma_{\infty }}\hat{\delta }\left(Q^{\prime }\right)+I_{\xi }\Theta^{\prime }\left(\phi ,\hat{\delta }\phi \right)=\int_{\partial \Sigma_{H}}\hat{\delta }\left(Q^{\prime }\right)+I_{\xi }\Theta^{\prime }\left(\phi ,\hat{\delta }\phi \right).$$
Remarkably, see [31], the relation above can be restated as the variation on shell, $\hat{\delta }$, at any of the boundaries of the generator of the diffeomorphism G(ξ). Therefore,
(23)$$\hat{\delta }{G\left(\xi \right)|}_{\partial \Sigma }=\int_{\partial \Sigma }\hat{\delta }g\left(\xi \right)=\int_{\partial \Sigma }\hat{\delta }\left(Q^{\prime }\right)+I_{\xi }\Theta^{\prime }\left(\phi ,\hat{\delta }\phi \right),$$
where ∂Σ stands for the asymptotic region or the horizon. In this way, in principle, one can compute (conserved) Hamiltonian charges g(ξ) by direct integration of Equation (Equation 23), provided the boundary conditions on each boundary hold.

On this point, it is necessary to comment on the difference between $\delta_{0}$, the functional variation, and $\hat{\delta }$. It is worth recalling that $\Theta^{\prime }\left(\phi ,\delta_{0}\phi \right)=0$ must be guarantied by the boundary conditions. Conversely, it is possible that $\Theta^{\prime }\left(\phi ,\hat{\delta }\phi \right)\neq 0$ because the boundary condition might not suffice. Fortunately, the vanishing of $\Theta^{\prime }\left(\phi ,\hat{\delta }\phi \right){|}_{\partial \Sigma }$ is not a requirement for Equation (Equation 23) to hold. A simple example of this occurs for GR for asymptotic flat spaces (Λ=0) under the usual boundary conditions. See, for instance [39], for a discussion.

The previous discussion became utmost relevant when it comes to the Killing vectors and its associated charges. In general, one is concerned only with the case where ξ stands for either time translation, rotations, or a linear combination of them that may define a Killing horizon on the space. It is direct to notice [40] that for rotations, the second term of Equation (Equation 23) vanishes identically, implying that Noether and Hamiltonian charges associated with rotational symmetries are the same. Conversely, for the time translation, the second term of Equation (Equation 23) may contribute, and thus, the Noether and Hamiltonian charges may differ in the case. This will be discussed in detail in the next section for gravity.

## 3. Lovelock Action Principle

In the previous section, both Hamiltonian and Noether charges have been identified and related. In this section, these results will be applied on Lovelock gravity—one of the simplest generalization of General Relativity in higher dimensions d>4. The Lovelock Lagrangian is the addition, with arbitrary coefficients $\left\{{\tilde{\alpha }}_{p}\right\}$, of the lower dimensional Euler densities [28,41]. The Lagrangian can be written as:(24)$$L=\sum_{p=0}^{\left[(d-1)/2\right]}{\tilde{\alpha }}_{p}R^{p}e^{d-2p}$$
where [(d−1)/2] is the integer part of (d−1)/2, and:(25)$$R^{p}e^{d-2p}=R^{a_{1}a_{2}}\wedge \ldots \wedge R^{a_{2p-1}a_{2p}}\wedge e^{a_{2p+1}}\wedge \ldots \wedge e^{a_{d}}\varepsilon_{a_{1}\ldots a_{d}}.$$
The variation of this action principle is given by:(26)$$\delta_{0}L=\sum_{p=0}^{\left[(d-1)/2\right]}p{\tilde{\alpha }}_{p}d\left(\delta_{0}\omega R^{p-1}e^{d-2p}\right)+{\mathrm{EOM}}_{e}\delta_{0}e+{\mathrm{EOM}}_{\omega }\delta_{0}\omega ,$$
where [41]:(27)$${\mathrm{EOM}}_{e}=\sum_{p=0}^{\left[(d-1)/2\right]}\left(d-2p\right){\tilde{\alpha }}_{p}R^{p}e^{d-2p-1}=0.$$
On the other hand, in general, ${\mathrm{EOM}}_{\omega }=0$ is satisfied by considering the Levi–Civita connection, i.e., if $T^{a}=de^{a}+\omega_{0\;b}^{a}\wedge e^{b}=0$ is satisfied. It is worth mentioning that for the Chern–Simons gravity [41], the Levi–Civita connection, though a solution, is not the most general solution to ${\mathrm{EOM}}_{\omega }$. From now on, the ∧-product will be omitted as its presence is self explanatory on the equations.

### 3.1. The Ground States and Regularization

The analysis of the asymptotic structure of Lovelock gravity solutions can be found, for instance, in [42]. Let us consider that ${\tilde{\alpha }}_{p}=0$ for p>I with [(n−1)/2]≥I≥1. Now, one can notice that the equations of motion can be written as:(28)$$G_{a_{d}}=\left(R^{a_{1}a_{2}}+\kappa_{1}e^{a_{1}}e^{a_{2}}\right)\ldots \left(R^{a_{2I-1}a_{2I}}+\kappa_{I}e^{a_{2I-1}}e^{a_{2I}}\right)e^{a_{2I+1}}\ldots e^{a_{d-1}}\varepsilon_{a_{1}\ldots a_{d}}=0,$$
where $\left\{\kappa_{i}\right\}$ is a set of constants to be determined from the set $\left\{{\tilde{\alpha }}_{p}\right\}$. Now, it is straightforward to notice that any space of constant curvature $\kappa_{i}$ is a solution of the EOM. These could be identified as the *ground states* of the theory, but this is not yet the final situation. By introducing a constant curvature ansatz $R^{ab}=xe^{a}e^{b}$, Equation (Equation 28) becomes $G_{a_{d}}=P_{l}\left(x\right)e^{a_{1}}\ldots e^{a_{d-1}}\varepsilon_{a_{1}\ldots a_{d}}$, where $P_{l}\left(x\right)$ is the polynomial:(29)$$P_{l}\left(x\right)=\sum_{p=0}^{I}{\tilde{\alpha }}_{p}x^{p}=\left(x+\kappa_{1}\right)\ldots \left(x+\kappa_{I}\right)=\prod_{i=1}^{I}\left(x+\kappa_{i}\right).$$
One can notice that the set $\left\{\kappa_{i}\right\}$ corresponds to the zeros of $P_{l}\left(x\right)$, which, in general, can be complex numbers with a non-null imaginary part. This restricts the number of potential ground states to be defined by $\left\{{\tilde{\alpha }}_{p}\right\}$ and can be called a dynamical selection of the ground sates. By the same token, one can assert that the positive or null $\kappa_{i}$ defines the possible asymptotic behaviors of the solutions of Equation (Equation 28), as those solutions must approach one of the ground states asymptotically. The case of a $\kappa_{i}<0$, which would correspond to a dS ground state, stands apart since there are no asymptotic regions in this case. It is worth mentioning that, as noticed in [43], in certain cases, the definition of a *ground state* can be extended to non-constant curvature spaces.

To proceed beyond the ground state solution, let us consider the case where $\kappa_{1}=\ldots =\kappa_{k}=l^{-2}$ in Equation (Equation 29), for some k≤I, with l∈R−{0}. $\kappa_{i}\neq l^{-2}$ for I≥i>k. This corresponds to the existence of a *k*-fold degenerate solution of $P_{l}\left(x\right)=0$. In this case, Equation (Equation 28) can be written as:(30)$${\left(R+\frac{e^{2}}{l^{2}}\right)}^{k}\left(\sum_{q=0}^{[(d-1)/2]-k}{\tilde{\beta }}_{q}R^{q}e^{d-2k-2q-1}\right)=0,$$
where the ${\tilde{\beta }}_{q}$ are arbitrary coefficients. The original ${\tilde{\alpha }}_{p}$ coefficients, mentioned above, can be written as:(31)$${\tilde{\alpha }}_{p}=\frac{1}{d-2p}\sum_{j=0}^{[(d-1)/2]-k}l^{2(j-k)}\left(\begin{array}{c} k\\ p-j \end{array}\right){\tilde{\beta }}_{j}.$$
Once this is explicit, one can realize that the respective associate family of solutions, satisfying Equation (Equation 30), must behave such that $\lim_{x\to \partial \Sigma_{\infty }}R^{ab}=-l^{-2}e^{a}e^{b}$.

### 3.2. Problems and a Solution

The Lovelock Lagrangian presents three problems. It is direct to confirm that the Lagrangian asymptotically becomes proportional to the element of volume of the space and therefore the action principle in Equation (Equation 24) diverges.

Second, by noticing that the boundary term is given by:(32)$$\Theta \left(\omega ,e,\delta \omega \right)=\delta \omega^{ab}\frac{\partial L}{\partial R^{ab}}=\sum_{p=0}^{\left[(d-1)/2\right]}p{\tilde{\alpha }}_{p}\delta \omega R^{p-1}e^{d-2p},$$
it is direct to realize that there is no proper set of boundary conditions under which this can vanish because asymptotically $\lim_{x\to \partial \Sigma_{\infty }}\delta \omega R^{p-1}e^{d-2p}\approx \delta \omega e^{d-2}$ and $e^{d-2}$ diverges.

Finally, one can notice that the Nöther current associated with the diffeomorphisms, x→x+ξ, which is given by [32]:(33)$${}^{*}J=-d\left(I_{\xi }\omega^{ab}\frac{\partial L}{\partial R^{ab}}\right),$$
where:(34)$$\frac{\partial L}{\partial R^{ab}}=\sum_{p=0}^{\left[(d-1)/2\right]}p{\tilde{\alpha }}_{p}\varepsilon_{abc_{1}\ldots c_{d-2}}R^{c_{1}c_{2}}\ldots R^{c_{2p-3}c_{2p-2}}e^{c_{2}p-1}\ldots e^{c_{d-2}},$$
becomes proportional to the spatial volume element, ${\left.e^{d-2}\right|}_{\Sigma }$, and thus it diverges as well.

These three problems can be solved simultaneously by the introduction of a regulator in the action principle [44]. In [44], for even dimensions, and later generalized in [37,38,45], a different method based on the addition of topological densities was introduced. This method is sketched in Appendix A.

The boundary conditions for an internal boundary, meaning the event horizon of a black hole $\partial \Sigma_{H}$, will be discussed in the sections. At the horizon, see Equation (Equation 32), is possible just fixing $\omega^{ab}$ as no divergences might come from $\partial L/\partial R^{ab}$. This naïve condition actually fixes the temperature [46] of the black hole since the surface gravity is defined by the second fundamental form of the horizon which, in turn, is the pull back of ω onto the horizon. In the next sections this will be discussed for a static geometry where the relation between fixing $\omega^{ab}$ and fixing the temperature of the horizon manifests.

## 4. Scale Transformations and an Improved Presymplectic Form

Before proceeding, it is worth it to comment on the EOM (Equation 30). In an AdS/CFT scenario, one can conjecture that somehow each different theory of gravity must correlate to a different conformal theory on the conformal infinity. Therefore, upholding the form of the EOM must be relevant in an AdS/CFT scenario as a way to be able to identify a single family of gravitational theories, given their common asymptotic form, see Equation (Equation 58) below, with a single family of would-be dual CFT living on the conformal infinity. On the other hand, it is direct to check that the independent variation of the coefficients ${\tilde{\alpha }}_{p}$ spoils the form of equations of motion (Equation 30). Because of that, in this work, the variation of the ${\tilde{\alpha }}_{p}$ would be constrained to maintain the form of Equation (Equation 24). This differs from most of the literature, see, for instance [14,15,47]. The difference arises because, even though the AdS radius *l* and the cosmological constant are connected, in order to maintain the form of Equation (Equation 30), each of the ${\tilde{\alpha }}_{p}$ in the Lovelock Lagrangian, with the exception of ${\tilde{\alpha }}_{1}$ the Newton constant, must vary along $l\to l+\hat{\delta }l$ and they must do it in a specify way. Obviously, this can be expected to render different results, and in particular, different definitions for thermodynamic pressure and volume.

In this work, it is assumed that *l* is a property of the theory; this differs from [48], where the cosmological constant in four dimensions is generated by a three-form or the most usual generation by the potential of a scalar field. In fact, the construction of the extended covariant space method below will treat *l* as an additional coordinate in the space parameters. In the same fashion, $\delta l^{-2}$ is to be considered an additional (co-)direction in the co-tangent space of the space of parameters. One can be concerned that *l*, from a physical point of view, not being a conserved charge, is essentially different from *M*, *J*, or the rest of the conserved charges, and yet is treated in a similar footing with them. From a mathematical point of view, and in principle, this is similar to the considered $l^{-2}$—an intensive quantity in thermodynamics where the conserved charges are the extensive variables.

Now that it has been established that the variation of *l* must preserve the EOM in Equation (Equation 31), it is necessary to implement how this will be done. Let us consider the infinitesimal global scale transformations:(35)$$e\to {\left(1+\sigma \right)}^{-1}e,$$
with |σ≪1|. It is straightforward to check that this preserves the EOM in Equation (Equation 31). These transformations can be reshaped, for convenience, into:(36)$$l\to l^{\prime }=\left(1+\sigma \right)l=l+\hat{\delta }l.$$
Now, by a direct (dimensional) analysis, one can notice that the coefficients ${\tilde{\alpha }}_{p}$, that give rise to Equation (Equation 30), not only must depend on *l* but they do it with different *powers* of *l*. Therefore, in order to vary them along $l\to l+\hat{\delta }l$, it is convenient to make an explicit dependency on *l* by defining a new set of coefficients $\left\{\alpha_{p}\right\}$ functionally independent of *l*. To clarify the analysis, let *L* be the unit of length, after fixing $c=\hbar =\kappa_{b}=1$. Notice that with these definitions, the action principle is dimensionless, $L^{0}$. By the same token, the units are given as follows: [energy (and enthalpy)] $=L^{-1}$, [entropy] = $L^{0}$, [temperature] = $L^{-1}$ and [force] = $L^{-2}$ in any dimension. Finally, [volume] $=L^{d-1}$ and [pressure = force/area] $=L^{-d}$, as expected.

The next consideration comes from recognizing the presence of quotient e/l in $R+{\left(e/l\right)}^{2}=0$ and realizing that $e^{a}/l$ is dimensionless. With this in mind, one can introduce a criterion to redefine the coefficients ${\tilde{\alpha }}_{p}$. One can notice that in four dimensions, the gravitational constant does not depend on the scale, and this can be extended to the corresponding Newton constant, defined by ${\tilde{\alpha }}_{1}$ (the constant accompanying the Ricci scalar in the Lovelock action), in any dimension. This imposes that $\alpha_{1}={\tilde{\alpha }}_{1}$ and is consistent with ${\tilde{\alpha }}_{1}$ having units $L^{2-d}$. $\left[{\tilde{\alpha }}_{1}\right]=L^{2-d}$ rules out any dependency of the standard gravitational *force* on *l*.

The dependency of the rest of the coefficients follows the same rule, and thus, in order to comply with the units, one must define ${\tilde{\alpha }}_{p}=l^{2p-2}\alpha_{p}$, where the $\alpha_{p}$ coefficients are functionally independent of *l* and satisfy $\left[\alpha_{p}\right]=L^{2-d}$∀p. This yields:(37)$$L=\sum_{p=0}^{\left[(d-1)/2\right]}l^{2p-2}\alpha_{p}R^{p}e^{d-2p}=l^{d-2}\sum_{p=0}^{\left[(d-1)/2\right]}\alpha_{p}\underset{L^{0}}{\underset{︸}{R^{p}{\left(\frac{e}{l}\right)}^{d-2p}}}.$$

After this analysis of the dependency on *l* of the coefficients, one can construct a pre-symplectic form that incorporates the $l\to l+\hat{\delta }l$ transformation. This was proposed in [30] for the four dimensional Einstein Hilbert action in similar terms. This yields,
(38)$$\hat{\delta }L=\sum_{p}l^{2p-2}p\alpha_{p}d\left(\hat{\delta }\left(\omega \right)R^{p-1}e^{d-2p}\right)+\left(2p-2\right)\alpha_{p}l^{2p-3}\hat{\delta }l\left(R^{p}e^{d-2p}\right).$$
Now, for notation, let us assume that last term is a total derivative, i.e., $\left(2p-2\right)\alpha_{p}l^{2p-3}\hat{\delta }l$$\left(R^{p}e^{d-2p}\right)=d\theta_{p}$. This is direct to prove, see below, for a static space. Therefore,
(39)$$\hat{\delta }L=\sum_{p}l^{2p-2}p\alpha_{p}d\left(\hat{\delta }\left(\omega \right)R^{p-1}e^{d-2p}\right)+d\theta_{p}.$$

Before concluding this subsection, it is worth mentioning that a transformation $l\to l+\hat{\delta }l$ induces at the boundary $R\times \Sigma_{\infty }$ a rigid Weyl transformation. However, considering a potential AdS/CFT interpretation, and the fact that a conformal structure should not be altered classically by this kind of transformation, then it would become necessary to promote $R\times \Sigma_{\infty }$ to a representative of a family of conformal manifolds, as defined, for instance, in [26].

### 4.1. Regularization in Even Dimensions

In d=2n dimensions, the regularization can be performed by adding the Euler density with an adequate coupling constant. This case is discussed in detail in [32] and sketched in Appendix A. The Lagrangian changes according to:(40)$$L\to L^{\prime }=l^{2n-2}\sum_{p=0}^{n-1}\alpha_{p}R^{p}{\left(\frac{e}{l}\right)}^{2(n-p)}+\alpha_{n}R^{n}$$
where:(41)$$\alpha_{n}=-\frac{l^{2n-2}}{n}\sum_{p=0}^{n-1}p\alpha_{p}{\left(-1\right)}^{n-p},$$
We must stress that in this case, regularization corresponds to the completion of the Lovelock polynomial, by including the Euler density with a very particular coupling constant $\alpha_{n}$. The corresponding improved Noether charge is given by the expression,
(42)$${}^{*}J^{\prime }=-d\left(I_{\xi }\omega^{ab}\frac{\partial L^{\prime }}{\partial R^{ab}}\right)\mathrm{and}Q^{\prime }=-I_{\xi }\omega^{ab}\frac{\partial L^{\prime }}{\partial R^{ab}}$$

Now, using this definition, the improved presymplectic form has the form:(43)$$\hat{\delta }\Theta^{\prime }\left(\phi ,\delta_{\xi }\phi \right)-\delta_{\xi }\Theta^{\prime }\left(\phi ,\hat{\delta }\phi \right)=\sum_{p=0}^{n}d\left(\hat{\delta }\left(Q_{p}\right)+I_{\xi }\left(l^{2p-2}\hat{\delta }\left(\omega \right)R^{p-1}e^{d-2p}+\theta_{p}\right)\right)$$
where:(44)$$Q_{p}=-l^{2p-2}p\alpha_{p}\left(I_{\xi }\omega \right)R^{p-1}e^{d-2p}.$$

A last relevant comment is in place. In Equation (Equation 43), one can observe the presence of:(45)$$I_{\xi }\left(\sum_{p=0}^{n}l^{2p-2}\hat{\delta }\left(\omega \right)R^{p-1}e^{d-2p}\right),$$
which vanishes by construction on $\partial \Sigma_{\infty }$ due to the (asymptotic) boundary conditions. See Appendix A for that construction.

### 4.2. Regularization in Odd Dimensions

The regularization of the Lovelock action for asymptotic AdS spaces in d=2n+1 dimensions differs from the even-dimensional case. The process of regularization in odd dimensions requires considering boundary terms that cannot be expressed in a closed from in terms of $R^{ab}$ and *e*. The regulator can be expressed, however, in terms of the second fundamental form one form $K^{i}$, which contains the *extrinsic curvature*, the intrinsic two form curvature of the boundary $R^{ij}$ as well as the pullback of the vielbein onto the boundary. For a discussion, see [37,38,45,49], and a review can be found in Appendix A. In this case, the regularized action principle is given by:(46)$$\begin{array}{ccc} I^{\prime } & = & \int_{M}l^{2n-2}\sum_{p=0}^{n}\alpha_{p}R^{p}{\left(\frac{e}{l}\right)}^{2(n-p)}\\ & + & \kappa \int_{\partial M_{\infty }}\int_{0}^{1}\int_{0}^{t}\left(Ke{\left(\tilde{R}+t^{2}{\left(K\right)}^{2}+s^{2}\frac{e^{2}}{l^{2}}\right)}^{n-1}\right)dsdt \end{array}$$
where:(47)$$\kappa =\frac{2l^{2n-2}}{n}\left(\sum_{p=0}^{n}p{\left(-1\right)}^{2n-2p}\alpha_{p}\right)\frac{\Gamma \left(n+\frac{1}{2}\right)}{\Gamma \left(n\right)\sqrt{\pi }}$$
with $\tilde{R}$ and *K* stand for the Riemann two-form and extrinsic curvature one-form respectively of the boundary $\partial M_{\infty }=R\times \partial \Sigma_{\infty }$.

In order to proceed, we must carefully discuss the variation, including the change of scale. This is given by:(48)$$\begin{array}{ccc} \hat{\delta }I^{\prime } & = & -\int_{\partial M_{H}}\sum_{p=0}^{n}\left(l^{2p-2}\hat{\delta }\left(\omega \right)R^{p-1}e^{2(n-p)+1}+\theta_{p}\right)\\ & + & \int_{\partial M_{\infty }}\sum_{p=0}^{n}\theta_{p}\\ & + & 2\kappa \left(n-1\right)\hat{\delta }l\int_{\partial M_{\infty }}\int_{0}^{1}\int_{0}^{t}\left(K\left(\frac{e}{l}\right){\left(\tilde{R}+t^{2}{\left(K\right)}^{2}+s^{2}\frac{e^{2}}{l^{2}}\right)}^{n-1}\right)dsdt\\ & - & 2\kappa \left(n-1\right)\hat{\delta }l\;\int_{\partial M_{\infty }}\int_{0}^{1}\int_{0}^{t}\left(K{\left(\frac{e}{l}\right)}^{3}{\left(\tilde{R}+t^{2}{\left(K\right)}^{2}+s^{2}\frac{e^{2}}{l^{2}}\right)}^{n-2}\right)dsdt\\ & + & \kappa \int_{\partial M_{\infty }}\int_{0}^{1}\left(e\hat{\delta }K-\hat{\delta }eK\right){\left(\tilde{R}+t^{2}{\left(K\right)}^{2}+t^{2}\frac{e^{2}}{l^{2}}\right)}^{n-1}dt \end{array}$$
On this point, it is good to stress that since the variation $\hat{\delta }$ includes variations along $\hat{\delta }l$, then $\left(e\hat{\delta }K-\hat{\delta }eK\right)\neq 0$. This will be fundamental for the computations.

## 5. Static Solution

The static solutions of Lovelock gravity of the form in Equation (Equation 30), see [24], can be written using the vielbein:(49)$$e^{0}=f\left(r\right)dt,e^{1}=f{\left(r\right)}^{-1}dr\mathrm{and}e^{i}=r{\tilde{e}}^{i},$$
where ${\tilde{e}}^{i}$ is the intrinsic vielbein for a constant curvature transverse section, Ω, which must be compact and closed. Therefore, the intrinsic curvature of the transverse section satisfies ${\tilde{R}}^{ij}=\gamma {\tilde{e}}^{i}{\tilde{e}}^{j}$ with γ a constant. Without loss of generality one can take γ=±1,0.

One can notice, see Equation (Equation 49), that the vielbein has been written such as r→∞ defines the asymptotic region $R\times \partial \Sigma_{\infty }$. Conversely, $f{\left(r\right)}^{2}=0$ defines an event horizon. The spin connections are given by:(50)$$\omega^{01}=\frac{1}{2}\frac{d}{dr}f{\left(r\right)}^{2}dt,\omega^{1i}=f\left(r\right){\tilde{e}}^{i}\mathrm{and}\omega^{ij}={\tilde{\omega }}^{ij}$$
where ${\tilde{\omega }}^{ij}$ is the intrinsic Levi-Civita spin connection defined from ${\tilde{e}}^{i}$. The curvatures are:(51)$$\begin{array}{ccc} R^{01}=-\frac{1}{2}\frac{d^{2}}{dr^{2}}f{\left(r\right)}^{2}dt\wedge dr, & & R^{0i}=-\frac{1}{2}\frac{d}{dr}f{\left(r\right)}^{2}fdt\wedge {\tilde{e}}^{i}\\ R^{1i}=-\frac{1}{2}\frac{d}{dr}f{\left(r\right)}^{2}f^{-1}dr\wedge {\tilde{e}}^{i} & \mathrm{and} & R^{ij}=\left(\gamma -f{\left(r\right)}^{2}\right){\tilde{e}}^{i}\wedge {\tilde{e}}^{j}. \end{array}$$

By using the ansatz in Equation (Equation 49) together with the time-like Killing vector $\xi =\partial_{t}$, one can show that:(52)$$\begin{array}{ccc} R^{p}e^{d-2p} & = & \frac{d^{2}}{dr^{2}}\left({\left(\gamma -f{\left(r\right)}^{2}\right)}^{p}r^{d-2p}\right)dt\wedge dr\wedge d\Omega \\ & = & -d\left(\frac{d}{dr}\left({\left(\gamma -f{\left(r\right)}^{2}\right)}^{p}r^{d-2p}\right)dt\wedge d\Omega \right)\\ \theta_{p} & = & -\left(2p-2\right)\alpha_{p}l^{2p-3}\hat{\delta }l\frac{d}{dr}\left({\left(\gamma -f{\left(r\right)}^{2}\right)}^{p}r^{d-2p}\right)dt\wedge d\Omega \\ I_{\xi }\omega R^{p-1}e^{d-2p} & = & \left(\frac{df{\left(r\right)}^{2}}{dr}{\left(\gamma -f{\left(r\right)}^{2}\right)}^{p-1}\right)r^{d-2p}d\Omega \\ \hat{\delta }\omega R^{p-1}e^{d-2p} & = & \hat{\delta }\left(\frac{df{\left(r\right)}^{2}}{dr}{\left(\gamma -f{\left(r\right)}^{2}\right)}^{p-1}\right)r^{d-2p}dt\wedge d\Omega \\ & + & \left(d-2p\right)\hat{\delta }\left(f{\left(r\right)}^{2}\right)\left({\left(\gamma -f{\left(r\right)}^{2}\right)}^{p-1}r^{d-2p}\right)dt\wedge d\Omega \end{array}$$
where $d\Omega =\varepsilon_{i_{1}\ldots i_{d-2}}{\tilde{e}}^{i_{1}}\wedge \ldots \wedge {\tilde{e}}^{i_{d-2}}$. Let us define for simplicity, Ω=∫dΩ as well.

With these results in mind one can evaluate Equation (Equation 43). First, one can notice that for $\xi =\partial_{t}$, $\theta_{p}$ is given by:(53)$$I_{\xi }\theta_{p}=-\left(2p-2\right)\alpha_{p}l^{2p-3}\hat{\delta }l\left(\frac{d}{dr}\left(\gamma -f{\left(r\right)}^{2}{)}^{p}r^{d-2p}\right)\right)d\Omega$$

### 5.1. Even Dimensions

In d=2n dimensions, the presymplectic form can be separated into two contributions from $\partial \Sigma_{\infty }$ and $\partial \Sigma_{H}$ that cancel each other. In this case, the construction is straightforward for both horizon and asymptotic region $\partial \Sigma_{\infty }$ and in both surfaces it is satisfied that:(54)$$\begin{array}{ccc} \Xi & = & \int_{\partial \Sigma }\sum_{p=0}^{n}\alpha_{p}\left(-p\left(2p-2\right)\hat{\delta }ll^{2-3}\left(\frac{d}{dr}f{\left(r\right)}^{2}\right){\left(\gamma -f{\left(r\right)}^{2}\right)}^{p-1}r^{d-2p}\right.\\ & - & l^{2p-2}p\hat{\delta }\left(\left(\frac{d}{dr}f{\left(r\right)}^{2}\right)\left({\left(\gamma -f{\left(r\right)}^{2}\right)}^{p-1}r^{d-2p}\right)\right)\\ & + & l^{2p-2}p\hat{\delta }\left(\left(\frac{d}{dr}f{\left(r\right)}^{2}\right){\left(\gamma -f{\left(r\right)}^{2}\right)}^{p-1}\right)r^{d-2p}\\ & - & \left.\left(2p-2\right)l^{2p-3}\hat{\delta }l\left(\frac{d}{dr}\left(\gamma -f{\left(r\right)}^{2}{)}^{p}r^{d-2p}\right)\right)\right)d\Omega . \end{array}$$
It is direct to notice that some formal simplifications occur; however, the explicit form depends on the boundary considered and its corresponding boundary conditions. Because of that, those simplifications will be carried out only after the boundary conditions are discussed in the next sections.

### 5.2. Odd Dimensions

In d=2n+1 dimensions, the Ξ at the horizon has exactly the form of Equation (Equation 54). The difference happens at the asymptotic region $\partial \Sigma_{\infty }$. See Appendix A. To proceed, it is necessary to work out the following set of relations in Equation (Equation 48)
$$\begin{array}{ccc} \theta_{p} & = & -\frac{d}{dr}\left({\left(\gamma -f{\left(r\right)}^{2}\right)}^{p}r^{d-2p}\right)dt\wedge d\Omega \\ \left(e\hat{\delta }K-\hat{\delta }eK\right){\left(\tilde{R}+t^{2}{\left(K\right)}^{2}+t^{2}\frac{e^{2}}{l^{2}}\right)}^{n-1} & = & tr\frac{d}{dr}{\left(\hat{\delta }\left(f^{2}\right)\left(\gamma +t^{2}\left(-f^{2}+\frac{r^{2}}{l^{2}}\right)\right)\right)}^{n-1}dt\wedge d\Omega \end{array}$$
(55)$$\begin{array}{ccc} K\frac{e^{3}}{l^{3}}{\left(\tilde{R}+t^{2}{\left(K\right)}^{2}+s^{2}\frac{e^{2}}{l^{2}}\right)}^{n-2} & = & \left(\left(\frac{df^{2}}{dr}\frac{r^{3}}{l^{3}}+6f^{2}\frac{r^{2}}{l^{3}}\right){\left(\gamma -t^{2}f^{2}+\frac{s^{2}r^{2}}{l^{2}}\right)}^{n-2}\right.\\ & + & \left.2f^{2}\frac{r^{3}}{l^{3}}\frac{d}{dr}{\left(\gamma -t^{2}f^{2}+\frac{s^{2}r^{2}}{l^{2}}\right)}^{n-2}\right)dt\wedge d\Omega \\ K\frac{e}{l}{\left(\tilde{R}+t^{2}{\left(K\right)}^{2}+s^{2}\frac{e^{2}}{l^{2}}\right)}^{n-1} & = & \left(\left(\frac{df^{2}}{dr}\frac{r}{l}+2\frac{f^{2}}{l}\right){\left(\gamma -t^{2}f^{2}+\frac{s^{2}r^{2}}{l^{2}}\right)}^{n-1}\right.\\ & + & \left.2f^{2}\frac{r}{l}\frac{d}{dr}{\left(\gamma -t^{2}f^{2}+\frac{s^{2}r^{2}}{l^{2}}\right)}^{n-1}\right)dt\wedge d\Omega \end{array}$$

### 5.3. Asymptotic Behavior

Following the discussion above, let us consider that the equations of motion have *k*-degenerated ground states of constant curvature $-l^{-2}$, i.e.,
(56)$$\frac{\partial L}{\partial e}=\frac{\partial L_{R}}{\partial e}=l^{d-3}{\left(R+\frac{e^{2}}{l^{2}}\right)}^{k}\left(\sum_{q=0}^{[(d-1)/2]-k}\beta_{q}R^{q}{\left(\frac{e}{l}\right)}^{d-2k-2q-1}\right)=0.$$
Here, $\beta_{q}=l^{d-2}{\tilde{\beta }}_{q}$ are arbitrary coefficients. One can notice that the EOM behaves asymptotically in the branch $\lim_{x\to \partial \Sigma_{\infty }}R^{ab}=-l^{2}e^{a}e^{b}$ as:(57)$$\lim_{x\to \partial \Sigma_{\infty }}\frac{\partial L}{\partial e}∼\left(\sum_{q=0}^{[(d-1)/2]-k}\beta_{q}{\left(-1\right)}^{q}\right)l^{d-3}{\left(R+\frac{e^{2}}{l^{2}}\right)}^{k}{\left(\frac{e}{l}\right)}^{d-2k-1}=0.$$
This implies that the solutions of this branch must behave asymptotically as:(58)$$\lim_{r\to \infty }f{\left(r\right)}^{2}∼\gamma +\frac{r^{2}}{l^{2}}-{\left(\frac{C}{r^{d-2k-1}}\right)}^{1/k},$$
where *C* is a constant to be determined from the exact solution. Remarkably, knowing this asymptotic behavior is enough to compute the variation of the asymptotic Nöther charges, Equation (Equation 33). However, as mentioned previously, one still has to concern about regularization of the action principle to obtain the proper Nöther charges.

### 5.4. Noether Charge in Even Dimensions

In even dimensions d=2n, the process of regularization is straightforward, see Appendix A. In the case at hand, the Killing vector $\xi =\partial_{t}$ defines the Killing horizon and the mass parameter.
(59)$$Q^{2n}\left(\partial_{t}\right)=\int_{\partial \Sigma_{\infty }}I_{\partial_{t}}\omega^{ab}\frac{\partial L^{\prime }}{\partial R^{ab}}=Cl^{2k-2}\left(\sum_{q=0}^{n-1-k}\beta_{q}{\left(-1\right)}^{q}\right)\Omega .$$
By identifying $M=Q\left(\partial_{t}\right)$ one can fix *C* such that:(60)$$M=Cl^{2k-2}\left(\sum_{q=0}^{n-1-k}\beta_{q}{\left(-1\right)}^{q}\right)\Omega ↔C=l^{2-2k}\frac{M}{\Omega }{\left(\sum_{q=0}^{n-1-k}\beta_{q}{\left(-1\right)}^{q}\right)}^{-1}.$$
It is direct to check that $\left[M\right]=L^{-1}$, as expected, while $\left[C\right]=L^{d-2k-1}$.

### 5.5. Noether Charge in Odd Dimensions

Certainly the most striking difference of the odd dimensional case, says d=2n+1, is the presence of an additional term corresponding to the *vacuum energy* of AdS${}_{2n+1}$. This has been obtained by several authors in different ways. See, for instance [49]. This vacuum energy, although its dependence on *l* is generic given *d*, it is not independent of the (Lovelock) gravitational theory considered. For the static case discussed above, it can be shown that this is given by:(61)$$Q^{2n+1}\left(\partial_{t}\right)=M+E_{0}\mathrm{with}E_{0}=\kappa \gamma^{n}\Omega ,$$
which is the result in [49] rewritten in the conventions of this work. To proceed, it will be useful to write down κ explicitly in this point, i.e.,
(62)$$E_{0}=\frac{2\Omega }{n}{\left(-1\right)}^{n+1}l^{2(n-1)}\gamma^{n}\frac{\Gamma \left(n+\frac{1}{2}\right)}{\sqrt{\pi }\Gamma \left(n\right)}\left(\sum_{p=0}^{n}p{\left(-1\right)}^{p}\alpha_{p}\right),$$
in order to make explicit the presence of the $\alpha_{p}$ coefficients in this expression. See Equation (Equation 31). As can be observed, $E_{0}$ depends on the particular Lovelock theory considered. It is also necessary to notice that:(63)$$C=l^{2-2k}\frac{M}{\Omega }{\left(\sum_{q=0}^{n-k}\beta_{q}{\left(-1\right)}^{q}\right)}^{-1},$$
which confirms, as previously, that $\left[M\right]=L^{-1}$ and $\left[C\right]=L^{d-2k-1}$.

### 5.6. Variation along the Space of Solutions

First, one must stress that the constant *C* is merely a function of the integration constants and therefore *C* lacks any physical meaning by itself. Conversely, the Noether and Hamiltonian charges are the physical meaningful quantities. In this way, *C* must be defined in terms of *M* and *l* to acquire a physical meaning.

One can notice that for the construction of the presymplectic form is necessary to consider the variation along *M* and *l*, and thus necessary to construct the variation of the conserved charges. In general, for the variation along *M*, the presence of $E_{0}$ is irrelevant. Conversely, the presence of $E_{0}$ for the variation along *l* is quite relevant.

The existence of Equations (Equation 60) and (Equation 61) is not necessary to compute the variation of the conserved charges, not even *M*, which, in this context, can be understood as the integral of $\tilde{\delta }M$. However, since Equations (Equation 60) and (Equation 61) actually fix the dependency of *C* on *l*, they can be considered shortcuts to compute $\hat{\delta }f{\left(r\right)}^{2}$.

### 5.7. Hamiltonian Variation

The variation of the Hamiltonian charges Equation (Equation 43) at the asymptotic region is given by:(64)$$\begin{array}{ccc} {\left.\hat{\delta }g\left(\partial_{t}\right)\right|}_{\partial \Sigma_{\infty }} & = & \lim_{r\to \infty }\sum_{p=0}^{\left[(d-1)/2\right]}\alpha_{p}\left(l^{2p-2}p\hat{\delta }f{\left(r\right)}^{2}{\left(\gamma +f{\left(r\right)}^{2}\right)}^{p-1}\left(d-2p\right)r^{d-2p-1}\right.\\ & - & \left.\left(2p-1\right)\left(d-2p\right){\left(\gamma -f{\left(r\right)}^{2}\right)}^{p}r^{d-2p-1}l^{2p-3}\hat{\delta }l\right)d\Omega . \end{array}$$
It is straightforward to notice that the form above can be casted as:(65)$${\left.\hat{\delta }g\left(\partial_{t}\right)\right|}_{\partial \Sigma_{\infty }}=\frac{\partial g}{\partial M}\hat{\delta }M+\frac{\partial g}{\partial l}\hat{\delta }l.$$
To compute each of the contributions one needs to separate the variation of $f{\left(r\right)}^{2}$ as:(66)$${\left.\hat{\delta }f{\left(r\right)}^{2}\right|}_{x\to \partial \Sigma_{\infty }}=\frac{\partial f{\left(r\right)}^{2}}{\partial M}\hat{\delta }M+\frac{\partial f{\left(r\right)}^{2}}{\partial l}\hat{\delta }l.$$
where $f{\left(r\right)}^{2}$ is given by Equation (Equation 58). It direct to compute the variation along $\hat{\delta }M$:(67)$$\begin{array}{ccc} {\left.\frac{\partial g}{\partial M}\right|}_{\partial \Sigma_{\infty }} & = & \lim_{r\to \infty }\sum_{p=0}^{\left[(d-1)/2\right]}\alpha_{p}\left(l^{2p-2}p\frac{\partial }{\partial M}f^{2}\left(r\right){\left(\gamma +f{\left(r\right)}^{2}\right)}^{p-1}\left(d-2p\right)r^{d-2p-1}\right)d\Omega \\ & = & \frac{1}{\Omega }d\Omega \end{array}$$
The variation along $\hat{\delta }l$ requires a careful discussion.

#### 5.7.1. For d=2n

In this case, the element to evaluate is given by:(68)$$\begin{array}{ccc} {\left.\frac{\partial g}{\partial l}\right|}_{\partial \Sigma_{\infty }} & = & \lim_{r\to \infty }\sum_{p=0}^{n}\alpha_{p}\left(l^{2p-2}p\frac{\partial }{\partial l}f^{2}\left(r\right){\left(\gamma +f{\left(r\right)}^{2}\right)}^{p-1}\left(d-2p\right)r^{d-2p-1}\right.\\ & - & \left.\left(2p-1\right)\left(d-2p\right){\left(\gamma -f{\left(r\right)}^{2}\right)}^{p}r^{d-2p-1}l^{2p-3}\right)d\Omega =0 \end{array}$$
and therefore the variation of the Hamiltonian charge corresponds to the variation of the Enthalpy,
(69)$${\left.\hat{\delta }G\left(\partial_{t}\right)\right|}_{\partial \Sigma_{\infty }}=\hat{\delta }M$$
as expected for d=2n.

#### 5.7.2. For d=2n+1

The computations in this case are cumbersome which requires us to consider the contribution of Equation (Equation 55) before taking the limit. Because of that, one can consider writing the expression above in terms of the Noether charge, meaning Equation (Equation 131). This yields,
(70)$${\left.\hat{\delta }G\left(\partial_{t}\right)\right|}_{\partial \Sigma_{\infty }}=\hat{\delta }\left(M+\kappa \gamma^{n}\Omega \right)-\int_{\partial \Sigma_{\infty }}I_{\xi }\Theta^{\prime }\left(\hat{\delta }e,\hat{\delta }\omega ,e,\omega \right)$$
where $\Theta^{\prime }\left(\hat{\delta }e,\hat{\delta }\omega ,e,\omega \right)$ was defined in Equation (Equation 48).

The explicit result will be discussed below for some relevant results.

### 5.8. The Horizon

To address the boundary conditions at the horizon one must reanalyze Equation (Equation 26). Unlike the asymptotic region, at the horizon, the simplest condition that ensures ${\left.\Theta \right|}_{\partial \Sigma_{H}}$ =0, Equation (Equation 26), since $\partial L/\partial R^{ab}$ is finite, is fixing ${\delta \omega |}_{\partial \Sigma_{H}}=0$. Now, considering the variation along the parameter of the solution in Equation (Equation 49), this is given by:(71)$$\hat{\delta }\omega^{ab}=\frac{1}{2}\delta_{01}^{ab}\hat{\delta }\left(\frac{d}{dr}f{\left(r\right)}^{2}\right)dt-\delta_{0i}^{ab}\hat{\delta }f\left(r\right){\tilde{e}}^{i}=0,$$
and thus both $f{\left(r\right)}^{2}$ and its derivative must be fixed along any trajectory in the space of parameters of solutions. Fixing the derivative of $f{\left(r\right)}^{2}$ corresponds to fixing the temperature. On the other hand, $\hat{\delta }f^{2}\left(r\right)=0$ is to be understood as the relation between the variations of the parameters of the solution, including horizon’s radius, such that for the new $r_{+}^{\prime }=r_{+}+\hat{\delta }r_{+}$$f^{2}\left(r_{+}^{\prime }\right)=0$ is satisfied. In a matter of speaking, this corresponds to promoting $f{\left(r_{+}\right)}^{2}\to f^{2}\left(r_{+},M,l,\ldots \right)=0$ subjected to:(72)$$\hat{\delta }f^{2}\left(r\right)=0=\frac{\partial }{\partial r_{+}}f^{2}\left(r\right)\hat{\delta }r_{+}+\frac{\partial }{\partial M}f^{2}\left(r\right)\hat{\delta }M+\frac{\partial }{\partial l}f^{2}\left(r\right)\hat{\delta }l+\ldots =0$$
This relation must be equivalent to the first law of the black hole thermodynamics defined by Equation (Equation 22). Otherwise, the thermodynamic evolution of the system would be inconsistent by having two different tangent vectors at each point.

Following with the construction, it is direct to evaluated Equation (Equation 54) subjected to $f{\left(r\right)}^{2}=0$ and $\hat{\delta }f{\left(r\right)}^{2}=0$. This yields:(73)$$\begin{array}{ccc} {\left.\hat{\delta }g\left(\partial_{t}\right)\right|}_{\partial \Sigma_{H}} & = & \sum_{p=0}^{\left[(d-1)/2\right]}\alpha_{p}\left(-l^{2p-2}p\left(\frac{d}{dr}f{\left(r_{+}\right)}^{2}\right)\gamma^{p}\hat{\delta }\left(r_{+}^{d-2p}\right)\right.\\ & + & \left.\left(\left(2p-2\right)\left(d-2p\right){\left(\gamma \right)}^{p}r_{+}^{d-2p-1}\right)l^{2p-3}\hat{\delta }l\right)d\Omega \end{array}$$
In Equation (Equation 73), one recognizes that the component along $\hat{\delta }r_{+}$ corresponds to the known expression for $T\hat{\delta }S$, where $T=1/\left(4\pi \right){\left(df{\left(r\right)}^{2}/dr\right)}_{+}$ [29]. This can be expressed as:(74)$$T\hat{\delta }S=T\left({\left(\gamma +\frac{r_{+}^{2}}{l^{2}}\right)}^{k-1}\left(\sum_{i=0}^{d-2k-1}\zeta_{i}\gamma^{p}r_{+}^{d-2(k+i+1)}\right)\right)\hat{\delta }r_{+}$$
where $\zeta_{i}$ are proportional to $\beta_{i}$ mentioned above in Equation (Equation 56). It is direct to show that in even and odd dimensions, see Appendix A, this is equivalent to the usual expression in [29,36]:(75)$$T\hat{\delta }S=T\hat{\delta }\left(2\pi \int_{\partial \Sigma_{H}}\frac{\partial L}{\partial R^{01}}\right).$$

The second term in Equation (Equation 73) corresponds to the generalization of the $V\hat{\delta }P$ term mentioned above. In this case, however, the connection with the cosmological constant and the *volume* of the black hole is not direct as for GR in four dimensions. For simplicity, this term will be called:(76)$$\begin{array}{ccc} w\hat{\delta }l & = & \sum_{p=0}^{\left[(d-1)/2\right]}\alpha_{p}\left(\left(2p-2\right)\left(d-2p\right){\left(\gamma \right)}^{p}r_{+}^{d-2p-1}\right)l^{2p-3}\hat{\delta }l\\ & = & \underset{∼V_{eff}}{\underset{︸}{\sum_{p=0}^{\left[(d-1)/2\right]}\alpha_{p}\left(\left(1-p\right)\left(d-2p\right){\left(\gamma \right)}^{p}r_{+}^{d-2p-1}\right)l^{2p}}}\underset{P}{\underset{︸}{\hat{\delta }\left(\;\frac{1}{l^{2}}\right)}}\\ & ∼ & V_{eff}dP \end{array}$$
It is interesting to compare this with the results obtained in [50]. In the language of our work, see Equation (Equation 76), the effective volume above has contributions coming from each of the terms in Lovelock Lagrangian, and therefore, the conjugate thermodynamic variable to the pressure *P* is constructed associated to all the terms in the Lovelock Lagrangian. This differs from [50], where only the Einstein Hilbert contribution is conjugate to their pressure *P* and the rest of the terms define additional conjugate variables. By the same token, this effective volume also differs from the one obtained in [51], as can be checked explicitly in the two examples displayed where the only contribution to the effective volume comes exclusively from the Einstein Hilbert term. Moreover, in [51] (the concept of) complexity, see [52], is used to perform the computation of the effective volume. We can see another different approach in [53], whose effective volume also differs from Equation (Equation 76).

One can notice that:(77)$$V_{eff}=\alpha_{0}dr_{+}^{d-1}\Omega +Correctionterms$$
meaning that this effective *volume* has corrections to the usual black hole volume ($∼r_{+}^{d-1}$) in powers of $r_{+}^{d-2p-1}l^{2p}$, due to the presence of higher curvature terms. It is worth mentioning that these corrections are such that p≠1 and d≠2p. Therefore, the Einstein Hilbert term with p=1 and the topological invariant terms with d=2p do not represent this type of correction. For the Einstein Hilbert theory (which consider p=0 and p=1), there are not corrections because only the p=0 term contributes to the effective volume, so this latter coincides with the usual definition of volume.

To make this more explicit, it is worth writing $V_{eff}$ to its full extension:(78)$$V_{eff}=\sum_{p=0}^{\left[(d-1)/2\right]}\sum_{j=0}^{[(d-1)/2]-k}\frac{1}{d-2p}\left(\begin{array}{c} k\\ p-j \end{array}\right)\beta_{j}\left(\left(1-p\right)\left(d-2p\right){\left(\gamma \right)}^{p}r_{+}^{d-2p-1}\right)l^{2p}$$
where $\beta_{j}$ are arbitrary coefficients. Equation (Equation 78) seems remarkable convoluted; however, the expression presents a large number of cancellations due to:(79)$$\left(\begin{array}{c} k\\ p-j \end{array}\right)=\frac{k}{k+j-p}B^{-1}\left(k+j-p\right)=0$$
for any (k+j−p)<1 integer.

To test the thermodynamic consequences of this result, some particular cases will be discussed in the next section.

### 5.9. Summary of First Law of Thermodynamics

From the analysis above, see Equations (Equation 69), (Equation 70), (Equation 73) and (Equation 76), it can be observed that in general the first law of thermodynamics presents contributions from infinity and from the horizon, yielding for even dimensions:(80)$$\hat{\delta }M=T\hat{\delta }S+V_{eff}\hat{\delta }P$$
and for odd dimensions:(81)$$\hat{\delta }M+\hat{\delta }\left(\kappa \gamma^{n}\Omega \right)-\int_{\partial \Sigma_{\infty }}I_{\xi }\Theta^{\prime }\left(\hat{\delta }e,\hat{\delta }\omega ,e,\omega \right)=T\hat{\delta }S+V_{eff}\hat{\delta }P$$
It must be stressed that the pressure can be defined consistently and universally as $\hat{\delta }P=\hat{\delta }\left(l^{-2}\right)$.

Furthermore, the horizon is modified by the scale change, thus, the change of scale introduces the effective volume $V_{eff}$ into the first law of thermodynamics. From Equation (Equation 77), $V_{eff}$ can be viewed as the usual definition of thermodynamics volume plus corrections due to the higher curvature terms. Thus, for the Einstein Hilbert theory, the effective volume coincides with the usual definition of volume.

In the next section, we show the form $\left(\kappa \gamma^{n}\Omega \right)-\int_{\partial \Sigma_{\infty }}I_{\xi }\Theta^{\prime }\left(\hat{\delta }e,\hat{\delta }\omega ,e,\omega \right)$ explicitly.

## 6. Relevant Cases

In this section, some relevant cases will be discussed.

### 6.1. Einstein in *d* Dimensions

Probably the simplest example of the previous construction is GR in d>3 dimensions. In this case, $\alpha_{0}$ and $\alpha_{1}$ are the only two non-null coefficients and they are fixed such that the EOM are given by:(82)$$\frac{\partial L}{\partial e}=\beta_{0}\left(R+\frac{e^{2}}{l^{2}}\right)\frac{e^{d-2}}{l^{d-1}}=0.$$
The static solution is defined by:(83)$$f{\left(r\right)}^{2}=\gamma +\frac{r^{2}}{l^{2}}-\frac{m}{r^{d-3}}$$
where $m=2M{\left(\Omega \beta_{0}\right)}^{-1}$ with $\hat{\delta }M$ the variation of the enthalpy. It is in fact direct to show that in this case the method yields:(84)$$\hat{\delta }M-M\hat{\delta }\ln\left(\frac{l}{\ell }\right)\delta_{d,2n+1}=T\hat{\delta }\left(\beta_{0}r_{+}^{d-2}\Omega \right)+\beta_{0}\left(r_{+}^{d-1}\Omega \right)\hat{\delta }\left(\frac{1}{l^{2}}\right).$$
The second term on the left is a novelty that requires specific discussion. First, it must be noticed that this additional term arises because of the vacuum energy in odd dimensions. On top of that, it is good to remember that the EOM only presents an approximated asymptotic (on-shell) AdS symmetry and thus one could speculate that this is connected with a failure of an exact AdS symmetry in odd dimensions. Although these ideas are quite compelling in the context of the AdS/CFT conjecture, where the vacuum energy indeed has clear interpretation, at this point, this is purely speculative thinking. A deeper analysis will be pursued in future works.

This new log term, in principle, modifies the usual thermodynamic evolution of the system. *ℓ* has been introduced just to provide a dimensionless expression in ln and represents a minimal radius for the possible AdS radii. One can read the usual entropy at the horizon,
(85)$$S∼\beta_{0}r_{+}^{d-2}\Omega ,$$
a pressure $P=l^{-2}$ and an effective volume given by $V_{eff}∼r_{+}^{d-1}$. The numerical factor can be fixed by the definition of the gravitational constant in *d* dimensions $\beta_{0}$.

### 6.2. Five Dimensional Einstein–Gauss–Bonnet Gravity

In [54], they restudied the static solution of the five-dimensional Lovelock gravity equations of motion:(86)$$l^{2}\left(5\alpha_{0}\frac{e^{4}}{l^{4}}+3\alpha_{1}R\frac{e^{2}}{l^{2}}+\alpha_{2}R^{2}\right)=l^{2}\left(R+\frac{e^{2}}{l^{2}}\right)\left(\beta_{1}R+\beta_{0}\frac{e^{2}}{l^{2}}\right)=0.$$
This solution was originally found in [55]. In Schwarzschild coordinates (see Equation (Equation 49)), this solution is defined by [54]:(87)$$f{\left(r\right)}^{2}=1+\frac{r^{2}}{4\alpha }-\frac{r^{2}}{4\alpha }\sqrt{1+\frac{16\alpha m}{r^{4}}+4\frac{\alpha \Lambda }{3}},$$
where the coefficient are given by:(88)$$\alpha =\frac{l^{2}\beta_{1}}{3!(\beta_{0}+\beta_{1})},\Lambda =-\frac{10\beta_{0}}{l^{2}\left(\beta_{0}+\beta_{1}\right)}\mathrm{and}m=\frac{M}{2(\beta_{0}+\beta_{1})\Omega },$$
These coefficients can be inverted into:(89)$$\beta_{0}=-\frac{\Lambda l^{2}}{5!\kappa^{2}}\mathrm{and}\beta_{1}=\frac{\alpha }{2l^{2}\kappa^{2}},$$
but restricted by $\beta_{0}+\beta_{1}={\left(12\kappa^{2}\right)}^{-1}$. Here, $\kappa^{2}=8\pi G$, with *G* the gravitational constant, as defined in [54]. In this case, the vacuum energy is given by [56]:(90)$$E_{0}=\frac{1}{8}l^{2}\beta_{0}\Omega$$
since the Noether charge is given by [57]:(91)$$Q\left(\partial_{t}\right)=M+\frac{1}{8}l^{2}\beta_{0}\Omega$$
The direct computation of Equation (Equation 70), in this case, yields:(92)$${\left.\hat{\delta }G\left(\partial_{t}\right)\right|}_{\partial \Sigma_{\infty }}=\hat{\delta }M+M\left(\frac{3\beta_{0}-7\beta_{1}}{\beta_{0}-\beta_{1}}\hat{\delta }\right)\ln\left(\frac{l}{\ell }\right),$$
where $\hat{\delta }M$ is to be considered as the variation of both the mass [56] but also of the enthalpy of the solution. As previously, *ℓ* has been introduced to have a dimensionless expression on ln and represents a minimal radius for the possible AdS radii. The presence of the variation of ln(l/ℓ), as mentioned above, can be argued is connected with the failure of a truly AdS symmetry in the bulk (the equations only have an approximate asymptotic on-shell AdS symmetry).

At the horizon, the presymptic of forms gives,
(93)$${\left.\hat{\delta }G\left(\partial_{t}\right)\right|}_{\partial \Sigma_{H}}=T\hat{\delta }S+V_{eff}\hat{\delta }P=T\left(\beta_{1}r_{+}^{2}+2\beta_{1}l^{2}+\beta_{0}r_{+}^{2}\right)\Omega \hat{\delta }r_{+}+\left(\left(\beta_{0}r_{+}^{4}\;+\beta_{1}l^{4}\right)\Omega \right)\hat{\delta }\left(\frac{1}{l^{2}}\right)$$
whose first term coincides with the usual Wald’s expression, Equation (Equation 75), for the entropy. Therefore, the first law of thermodynamics in this case is given by:(94)$$\hat{\delta }M+M\left(\frac{3\beta_{0}-7\beta_{1}}{\beta_{0}-\beta_{1}}\right)\hat{\delta }\ln\left(\frac{l}{\ell }\right)=T\hat{\delta }S+\underset{V_{eff}}{\underset{︸}{\left(\left(\beta_{0}r_{+}^{4}\;+\beta_{1}l^{4}\right)\Omega \right)}}\hat{\delta }\left(\frac{1}{l^{2}}\right)$$
The existence of $V_{eff}$ in this case can be considered as a contribution due to the change acting on the horizon. This correction, however, differs from the volume computed for generic Lovelock theories found in [58]. This effective volume can be understood as a type of Van der Waals corrections to the volume. The definition of the pressure as $P=l^{-2}$ is feature that will be generic for the rest of the examples.

### 6.3. Born-Infeld

In this case d=2n and the Lagrangian, once the regulator is added, has the form of a perfect binomial:(95)$$L_{R}=\beta_{0}l^{2n-3}{\left(R+\frac{e^{2}}{l^{2}}\right)}^{n},$$
and the EOM are:(96)$$l^{2n-5}\beta_{0}{\left(R+\frac{e^{2}}{l^{2}}\right)}^{n-1}e=0$$
The solution in this case is defined by:(97)$$f{\left(r\right)}^{2}=\gamma +\frac{r^{2}}{l^{2}}-{\left(\frac{m}{r}\right)}^{\frac{1}{n-1}}$$
By identifying the Noether charge by:(98)$$Q(\partial_{t})=M=H$$
From this, it is direct to check explicitly that $T\hat{\delta }S$ coincides with the definition in Equation (Equation 75). The $w\hat{\delta }l$ is given in this case by:(99)$$w\hat{\delta }l=-\beta_{0}l^{2n-3}r_{+}{\left(\gamma +\frac{r_{+}^{2}}{l^{2}}\right)}^{n-2}\left(\left(n-2\right)\gamma -\frac{r_{+}^{2}}{l^{2}}\right)\hat{\delta }\left(\frac{1}{l^{2}}\right)$$
Once again, in this case, one can take Equation (Equation 99) in the form $V\hat{\delta }P$ with $P=l^{-2}$ by defining the effective volume,
(100)$$V_{eff}=\beta_{0}l^{2n-3}r_{+}{\left(\gamma +\frac{r_{+}^{2}}{l^{2}}\right)}^{n-2}\left(\frac{r_{+}^{2}}{l^{2}}-\left(n-2\right)\gamma \right).$$
In this case, one can be confused because the effective volume becomes proportional to volume for $r_{+}\gg l$, but this is not an adequate limit.

### 6.4. Pure Lovelock

Pure Lovelock theory corresponds to just considering a single term in the Lovelock series plus the term associated with $\alpha_{0}$, meaning the cosmological constant. The EOM in this case can be cast in the form,
(101)$$l^{2s-3}\gamma_{s}\left(R^{s}\pm {\left(\frac{e}{l}\right)}^{2s}\right)e^{d-2s-1}=0,$$
where $\alpha_{s}={\left(d-2s\right)}^{-1}\gamma_{s}$ and $\alpha_{0}=d^{-1}\gamma_{s}$, and therefore $\gamma_{s}$ is an adjustment for $\alpha_{0}$.

As can be observed, this is an interesting example of how the Lovelock action gives rise to solutions, see Equation (Equation 101), behaving for r→∞ such as Schwarzschild solutions. This only feature makes Pure Lovelock gravity remarkably interesting. Let us recall that the case of interest has a ground state satisfying $R+e^{2}/l^{2}=0$. The double sign ± in Equation (Equation 101) comes from the fact that, depending on *s* being an even or odd integer, either positive or negative cosmological constant could give rise to solutions with an AdS asymptotic region. The exact static solution for odd s=2h+1, which corresponds to negative cosmological constant, can be written as:(102)$$f{\left(r\right)}_{odd}=1+\frac{r^{2}}{l^{2}}{\left(1-\frac{m}{r^{d-1}}\right)}^{\frac{1}{2h+1}}$$
whose asymptotic form is given by:(103)$$\lim_{r\to \infty }f{\left(r\right)}_{odd}\approx 1+\frac{r^{2}}{l^{2}}-\frac{1}{2h+1}\frac{m}{l^{2}r^{d-3}}.$$
with m>0. On the other hand, for even s=2h:(104)$$f{\left(r\right)}_{odd}=1+\frac{r^{2}}{l^{2}}{\left(1+\frac{m}{r^{d-1}}\right)}^{\frac{1}{2h}}$$
and the asymptotic form is given by:(105)$$f{\left(r\right)}_{even}\approx 1+\frac{r^{2}}{l^{2}}+\frac{1}{2h}\frac{m}{l^{2}r^{d-3}}.$$
with m>0. Because of this and the lack of horizon in this case, only the thermodynamics of odd *s* can be explored.

#### 6.4.1. Even Dimensions with s Odd

In even dimensions, let us say that d=2n with n≥2, there is no vacuum energy and thus the only contribution to the first law of thermodynamics arising from the horizon. For odd s=2h+1 the Noether charge is given by:(106)$$m_{s=2h+1}^{d=2n}=\frac{M}{\left(n-1\right)\Omega \gamma_{2h+1}}$$
where one can notice that this expression is independent of *h*. This is due to the contributions from the conformal infinity that must correspond to those of the k=1 (Einstein gravity). At the horizon, the rest of the first law of thermodynamics is given by:(107)$${\left.\hat{\delta }G\left(\xi \right)\right|}_{H}=T\hat{\delta }S+V_{eff}\hat{\delta }P$$
where the entropy and the effective volume are given by:(108)$$\begin{array}{ccc} S & = & \alpha_{2h+1}\left(2h+1\right){\left(\frac{r_{+}^{2}}{l^{2}}\right)}^{n-2h-1}\\ V_{eff} & = & \alpha_{0}\left(2n\right)r_{+}^{2n-1}+\alpha_{2h+1}\left(2h\right)\left(4h+2-2n\right)r_{+}^{2n-4h-3}l^{4h+2} \end{array}$$
where, as mentioned above, $\alpha_{s}={\left(d-2s\right)}^{-1}\gamma_{s}$ and $\alpha_{0}=d^{-1}\gamma_{s}$. The expression for the entropy is given by the Wald expression [29]. One can notice that the reason for the correction term in the effective volume is the presence of the higher power of the Riemann tensor in the action principle. The correction term is new respect to the volume computed under a variation of parameters in reference [59].

#### 6.4.2. Odd Dimension with s Odd

In this case, let us consider that d=2n+1. In this case, the contribution from infinity is given by:(109)$${\left.\hat{\delta }G\left(\xi \right)\right|}_{\infty }=\hat{\delta }M+M\hat{\delta }\ln\left(\frac{l}{\ell }\right).$$
On the other hand, from the horizon, ${\left.\hat{\delta }G\left(\xi \right)\right|}_{H}=T\hat{\delta }S+V_{eff}\hat{\delta }P$, where the entropy and volume takes the explicit form:(110)$$\begin{array}{ccc} S & = & \alpha_{2h+1}\left(2h+1\right){\left(\frac{r_{+}}{l}\right)}^{2n-4h-1}\\ V_{eff} & = & \alpha_{0}\left(2n+1\right)r_{+}^{2n}+\alpha_{2h+1}\left(2h\right)\left(4h+1-2n\right)r_{+}^{2n-4h-2}l^{4h+2} \end{array}$$
Therefore, the new first law for this case is represented by:(111)$$\hat{\delta }M=T\hat{\delta }S+V_{eff}\hat{\delta }P-M\hat{\delta }\ln\left(\frac{l}{\ell }\right)\delta_{d,2n+1}$$

### 6.5. Chern–Simons Gravity

From the point of view of the equations above, this case merely corresponds to the case in odd dimensions, d=2n+1, with k=n. However, in this case, the action principle becomes invariant under the larger local AdS transformation, instead of only Lorentz transformations, see, for instance [41]. In this case, *m* is given by:(112)$$m=\frac{M}{\beta_{0}l^{2}2\left(n-1\right)\Omega }$$
where $\beta_{0}$ is a global constant to fix. The vacuum energy [49] is given by $E_{0}=-l^{2n-2}\beta_{0}\gamma^{n}$ as expected.

The variation of conserved charges at the conformal infinity is given by:(113)$${\left.\hat{\delta }G\left(\xi \right)\right|}_{\infty }=\hat{\delta }M.$$
Here, we can notice the absence of any contribution related with a change of scale. One can speculate that it is because of the local AdS symmetry of the action principle. The contribution from the horizon is given by:(114)$${\left.\hat{\delta }G\left(\xi \right)\right|}_{H}=T\left[-n\beta_{0}\gamma {\left(r_{+}^{2}+\gamma l^{2}\right)}^{n-1}\right]\hat{\delta }r_{+}+\beta_{0}\left(r_{+}^{2}-\left(n-1\right)\gamma l^{2}\right){\left(r_{+}^{2}+\gamma l^{2}\right)}^{n-1}\hat{\delta }\left(\frac{1}{l^{2}}\right),$$
where we can recognize the known value of the entropy for Chern–Simons. The second contribution corresponds to the $V_{eff}dP$ term with:(115)$$V_{eff}=\beta_{0}\left(r_{+}^{2}-\left(n-1\right)\gamma l^{2}\right){\left(r_{+}^{2}+\gamma l^{2}\right)}^{n-1}$$

## 7. Conclusions and Prospects

In this work, we have studied the first law of thermodynamics in an alternative way, so we have explored the consequences of scale transformations, which preserve the form of the equations of Lovelock gravity whose solutions are asymptotic AdS spaces. This change of scale have been expressed in terms of changes of the corresponding AdS radius, *l*. This transformation introduces two additional terms in the first law of black hole thermodynamics. One comes from the horizon and defines an effective volume $V_{eff}$ and pressure $P\approx l^{-2}$. The second one from the conformal infinity in odd dimensions is proportional to $\hat{\delta }\ln\left(l\right)$.

With respect to the $V_{eff}dP$ term, its origin is clear. The horizon is modified by any change of scale, and thus, an additional term in the first law of thermodynamics must arise to compensate accordingly. However, it must be stressed that the universal $P∼l^{-2}$, attained in this work is due to the formalism introduced and the condition that the form Equation (Equation 30) be maintained. Conversely, $V_{eff}$ depends on the theory considered, and coincides only with the usual definition of thermodynamic volume for the Einstein Hilbert theory. One can interpret the effective volume as the EH (thermodynamic) volume plus corrections due to the higher curvature terms, in complete analogy with the usual interpretation of the entropy as the EH entropy plus corrections due to the higher curvature terms presented. See, for instance [60]. It is interesting to compare our results with those found in the literature based on variations of the cosmological constant. It is direct to notice that even though a term VdP is obtained, the corresponding results for *P* and $V_{eff}$ can differ. This potential difference can be tracked back to the variation of scale that was set by preserving the form of Equation (Equation 30). To address the consequences, this difference requires a thorough analysis of the thermodynamics evolution of this black hole. Phase transitions, lsuch as the Hawking–Page one, are interesting problems that will be addressed in future works.

A second additional correction to the first law arises for any Lovelock theory, but Chern Simons, in odd dimensions with the form ∼$M\hat{\delta }\ln\left(l/\ell \right)$. The first thing to notice, and emphasize, is that this is a general result independent of the theory considered and only absent in Chern Simons.

The absence of the ∼ln(l/ℓ) term for Chern Simons gravity allows us to speculate about the meaning of the additional term for the rest of generic Lovelock theories in odd dimensions. Unlike the rest of the Lovelock theories, Chern Simons gravity is a gauge theory, in this case for the AdS group, and this enlarged local symmetry modifies the physical meaning of the scale transformations in the bulk. For the rest of the Lovelock theories, a local AdS symmetry only could emerge as an approximate on shell locally asymptotic symmetry, and thus, one could speculate that the addition term is connected with the failure of an exact AdS symmetry. This larger local symmetry imposes additional constraints to be satisfied that, in turn, could be restricting the variation of the vacuum energy such that its contribution to the first law of thermodynamics vanishes. Unfortunately, neither the method developed in this work, nor the form of $M\hat{\delta }\ln\left(l/\ell \right)$, seems to provide enough information to confirm any of this. Certainly, additional study, beyond a purely thermodynamics framework, is required to establish a concrete connection between local symmetries and the absence of the additional term.

As mentioned, this is a different approach in at least three different ways. The coupling constant are varied such that the asymptotia be preserved. The thermodynamics is explored by an improved version of the phase space analysis. These two fundamental differences, we believe, are responsible for the arise of the new term. Another fundamental difference, any other method does not considered variation along the regularized conserved charges. For instance, Komar’s integrals are fundamentally divergent for ALAdS spaces and need a regularization scheme to become finite. It is good to mention that the conserved charges used in this work are actually generalization of the usual Komar’s integrals, in the sense that they are Noether charges. For example, the regularized method mentioned in [50] differs from those that gave rise to the conserved charges mentioned in our work.

Furthermore, it is worth mentioning that the logarithmic term has a dependence on the parameter *l*, which is a parameter of the equations of motion. Such dependence arises of the variation of the vacuum energy, as we can see in Equation (Equation 70), where, following the definitions of conserved charges used in this work, see for example [56,61], the vacuum energy has a dependence on the parameters of the equations of motion, which coincide with *l* for the Einstein Hilbert theory.

Before finishing, let us comment on some concrete prospects of this work. For Lovelock theories, we have studied phase transitions in the extended phase space in several works using different techniques, see, for example [16,62]. In doing that, it has been obtained that their phase transitions are analogues to liquid/gas transitions in Van der Waals theory. In reference [63], it was conjectured that in Lovelock theories, there could be *n-tuple critical points*. This seems to be confirmed in [16], where, using results obtained in [50], they obtained multiple critical points for charged solutions. In reference [64], it is argued that the values of the critical exponents, for Lovelock gravity, can differ from those of a Van der Waals gas. Now, in this still very open scenario, a natural next step is the analysis of phase transitions in our framework. This is particularly relevant since a different expression for the effective volume and a modified first law of thermodynamics have been obtained. In particular, a couple of very relevant questions are raised by our results. First, are the phase transitions still analogous to liquid/gas transitions in Van der Waals theory? Further, does the number of critical points, with respect to the previous results, increase or decrease? Finally, it seems quite interesting to reassert the Hawking–Page phase transition [15], given the modified thermodynamics obtained in this work.

## Data Availability

Not applicable.

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
