# Peer review of "An Alternative Study about the Geometry and the First Law of Thermodynamics for AdS Lovelock Gravity, Using the Definition of Conserved Charges"

_entropy, 2022, doi:10.3390/e24091197_

Round 1

Reviewer 1 Report

   In this paper, authors proposed an alternative study about the geometry and the first law of thermodynamics in the extended phase space for AdS Lovelock gravity.
    I find that the manuscript is interesting but I advise the authors to make some revisions:
    1). In page 34, the sentence in the Line 5 of the paragraph 2 possesses a spelling mistake
“…considered variation a long the regularized.
    2) In Section
VI, the authors recovered that a second additional correction to the first law arises for any Lovelock theory, but Chern Simons in odd dimensions. However, this important point does not appear in the abstract. I suggest the authors update the abstract.
    3). This manuscript is less verbose. For instance, in the part of
Change of the cosmological constant in four dimensions and generalizations”, the authors discussed the Schwarzschild-AdS spaces. However, there is too much detail of discussions for the horizon structures of Schwarzschild-AdS spaces. I suggest that authors focus on the conserved charges and the first law of thermodynamics.

     In my opinion it can be published in Entropy if the authors revised the
manuscript following the above suggestions
.

Author Response

Response to Reviewer 1

We would like to thank the referee for his/hers comments.

    1). In page 34, the sentence in the Line 5 of the paragraph 2 possesses a spelling mistake “…considered variation a long the regularized….

Answer:

We would like to thank the referee for calling our attention to these typos. We are really sorry about them. we have amended to text and now we expect the text to be free from typos.

    2) In Section VI, the authors recovered that a second additional correction to the first law arises for any Lovelock theory, but Chern Simons in odd dimensions. However, this important point does not appear in the abstract. I suggest the authors update the abstract.

Answer:

Following the recommendation, we have included an explicit mention of the Chern Simons case in the abstract.

    3). This manuscript is less verbose. For instance, in the part of Change of the cosmological constant in four dimensions and generalizations”, the authors discussed the Schwarzschild-AdS spaces. However, there is too much detail of discussions for the horizon structures of Schwarzschild-AdS spaces. I suggest that authors focus on the conserved charges and the first law of thermodynamics

Answer:

Following this recommendation, we have removed from the text the discussion, in the section "change..", what, in our opinion, was not directly related to the subsequent discussion in terms of the new formalism developed.

Reviewer 2 Report

Report on the paper entitled as “An alternative study about the geometry and the first law of thermodynamics for AdS Lovelock gravity, using the definition of conserved charges” by M. Estrada, R. Aros, and P. Pereira submitted for publication to the Journal “Entropy”.

 In the paper a new view on the first law of thermodynamics of black holes in the extended phase space for AdS Lovelock gravity is developed. As a part, scale transformations in presymplectic form, in order to derive this law, are considered. The new form for the first law of thermodynamics coincides with the standard form of one in 4 dimensions in general relativity. When higher dimensions in Lovelock gravity are under consideration, the variation of the Noether charge initiates an appearance of two new terms. The first of them corresponds to corrections of the usual definition of thermodynamics volume at the horizon, due to the presence of the higher curvature terms, and defines an effective volume and pressure. The analogous term has been obtained in other approaches, and future studies and comparisons are planned. The second one arises from the asymptotic region in the case of odd dimensions and corresponds to a contribution due to scale transformation that essentially depends on the ratio of the AdS radius to a scale. One has to note that this is a general result independent of the theory considered and only absent in Chern-Simons theory.

 The paper presents a fundamental study of the pressing and important issue concerning important sides of the problem. The definitions of the Noether charges related to diffeomorphisms and the outline of the phase space are given in detail. The elements of the Lovelock theory are represented in the convenient for the study form, necessary notions of a regularization technique are given. Various possibilities solutions of the Lovelock theory are considered, and various illustrative and useful applications have been provided.

At last, the topic of the paper quite corresponds to the direction of publications on the Journal “Entropy”. The language is good, the presentation is clear and consequent.

 Keeping all the above in mind I recommend to publish the paper “An alternative study about the geometry and the first law of thermodynamics for AdS Lovelock gravity, using the definition of conserved charges” by M. Estrada, R. Aros, and P. Pereira in the Journal “Entropy” in the present form.

Author Response

We would like to thank the referee for his/her kind comments.